# Evaluation of Colon-Specific Plasma Nanovesicles as New Markers of Colorectal Cancer

**DOI:** 10.3390/cancers13153905

**Published:** 2021-08-03

**Authors:** Inga Nazarova, Maria Slyusarenko, Elena Sidina, Nadezhda Nikiforova, Vladislav Semiglazov, Tatiana Semiglazova, Achim Aigner, Evgeny Rybakov, Anastasia Malek

**Affiliations:** 1Sub-Cellular Technology Lab., N.N. Petrov National Medical Center of Oncology, Department of Innovative Methods of Therapeutic Oncology and Rehabilitation, 197758 St. Petersburg, Russia; slusarenko_masha@mail.ru (M.S.); sidina@mail.ru (E.S.); niki2naden_ka@mail.ru (N.N.); vsemiglazov@mail.ru (V.S.); tsemiglazova@mail.ru (T.S.); anastasia@malek.com (A.M.); 2Department of Oncology, Pavlov First Saint Petersburg State Medical University, 197022 St. Petersburg, Russia; 3Clinical Pharmacology, Rudolf-Boehm-Institute for Pharmacology and Toxicology, Faculty of Medicine, University of Leipzig, 04107 Leipzig, Germany; achim.aigner@medizin.uni-leipzig.de; 4Ryzhikh National Medical Research Center of Coloproctology, Department of Oncoproctology, 123423 Moscow, Russia; erybakov@gmail.com; 5Onco-System Co., Ltd., 143026 Moscow, Russia

**Keywords:** extracellular nanovesicles, epithelial differentiation, colon cancer

## Abstract

**Simple Summary:**

High mortality in patients with colorectal cancer (CRC) is one of the main problems in healthcare. This is due to the use of invasive and existing noninvasive screening methods whose resources are limited. A promising alternative is a study of circulating in plasma extracellular nanovesicles (ENVs) reflecting the same composition of biomarkers with the secreted cells, forming subpopulations of tissue-specific ENVs. During the work, we have selected potential colon biomarkers from databases. The study aims to develop a quantitative method for isolating colon-specific ENVs based on the formation of immunocomplexes «beads with antibodies to specific biomarkers». We found that the amount of ENVs carrying potential colon biomarkers was higher in the patients with an IV CRC-stage compared with the healthy donors. It showed a high coefficient of diagnostic significance of these biomarkers in the CRC prognosis. These results will give an impetus to a deeper study of the ENVs as identifiers of cancer’s development.

**Abstract:**

Purpose: Developing new and efficient approaches for the early diagnosis of colorectal cancer (CRC) is an important issue. Circulating extracellular nanovesicles (ENVs) present a promising class of cancer markers. Cells of well-differentiated adenocarcinomas retain the molecular characteristics of colon epithelial cells, and the ENVs secreted by these cells may have colon-specific surface markers. We hypothesize that an increase in the number of ENVs carrying colon-specific markers could serve as a diagnostic criterion for colorectal cancer. Experimental design: Potential colon-specific markers were selected based on tissue-specific expression profile and cell surface membrane localization data. Plasma was collected from CRC patients (n = 48) and healthy donors (n = 50). The total population of ENVs was isolated with a two-phase polymer system. ENVs derived from colon epithelium cells were isolated using immune-beads with antibodies to colon-specific markers prior to labelling with antibodies against exosomal tetraspanins (CD63 and CD9) and quantification by flow cytometry. Results: The number of ENVs positive for single colon cancer markers was found to be significantly higher in the plasma of CRC patients compared with healthy donors. The efficacy of detection depends on the method of ENV labelling. The diagnostic efficacy was estimated by ROC analysis (the AUC varied between 0.71 and 0.79). The multiplexed isolation of colon-derived ENVs using immune-beads decorated with antibodies against five markers allowed for a further increase in the diagnostic potency of the method (AUC = 0.82). Conclusions: ENVs derived from colon epithelium may serve as markers of differentiated CRC (adenocarcinomas). The composition of ligands used for capturing colon-derived ENVs and their method of labelling are critical for the efficacy of this proposed diagnostic approach.

## 1. Introduction

The concept of the “liquid biopsy” is a not new; however, it is still an attractive alternative to traditional methods of cancer diagnostics. Different liquid biopsy platforms are available and are at different stages of clinical implementation. These include the assessment of free-circulating plasma proteins or glycoproteins, circulating cell-free nucleic acids, extracellular nanovesicles (ENVs), and circulating tumor cells (CTC).

Extracellular vesicles are an extensive heterogeneous group of membrane-covered vesicles, such as microvesicles, apoptotic bodies, and nanovesicles (exosomes), arising through the mechanism of endosomal transport of all cell types and release into the external space [1,2].

The biogenesis of these types of vesicles differs. Microvesicles with sizes 150–1000 nm [1] are formed by budding outward from the plasma membrane [2], while the formation of nanovesicles with sizes of 50–150 nm [1] takes place in several stages [2,3]: (i) the previous stage, with invagination of the plasma membrane’s domains, covered with clathrin (clathrin-coated vesicles (CCVs)); (ii) work of the ESCRT (Endosomal Sorting Complex Required for Transport) machinery, consisting of four functional complexes (ESCRT-0, -I, -II, and -III), which are at different stages involved in the sorting of ubiquitinated cargo and contributes to the development CCVs in an early-endosome (EE) carrying ubiquitinating cargo; (iii) the secondary invagination of EEs, forming intraluminal vesicles (ILVs) that accumulate and mature into multivesicular bodies; and (iv) some multivesicular bodies delivering ubiquitinated membrane proteins for their degradation in lysosomes while others release ILVs into the extracellular space, now called nanovesicles (exosomes) [4,5].

There are also some ECSRT-independent pathways for the budding and release of nanovesicles. Some studies have described that the formation of ILVs in oligodendrocyte precursor’s cells (Oli-neu cells) require sphingolipid ceramide [6]. There is also a ceramide dependent pathway. A second pathway is a tetraspanin-dependent pathway: a clustering of CD63 during the formation of ILVs in melanocytes under the conditions of depletion of components of ESCRT machinery [7].

The content of EVs is characterized by the presence of proteins included in membrane transport and fusion: Rab; GTPases; annexins; flotillin; tetraspanins CD63, CD81, and CD9; chaperons (HSP70 and 90); lipid rafts; and microRNAs [8].

However, a clear boundary between the subtypes of EV nanovesicles and microvesicles has not yet been found. It all depends on the method of vesicle isolation and the purpose of the study [9]. Extracellular vesicles take part in various physiological and pathological processes [10].

Since considerable progress has been made in understanding the biogenesis, structure, and functions of ENVs [11], circulating ENVs have attracted great attention as promising cancer markers. 

The most popular approach in ENV-based cancer diagnostics is the quantification of so-called tumor-derived vesicles. Already in 2015, the profiling of ENVs positive for membrane-associated lung cancer-related proteins was shown to allow for accurate distinction between advanced non-small lung cancer patients and healthy donors [12]. Later, other surface components (Glypican-1, PD-L1, and CSPG4) were explored as cancer cell ENV markers as well [13,14,15] and promising results stimulated further exploration of tumor-derived ENVs as cancer markers [16]. However, this strategy requires confident knowledge of cancer-specific ENV membrane-associated markers, and the relative specificity of such markers compromises the diagnostic specificity of this method as a whole. An alternative approach to ENV-based cancer diagnostics is the assessment of cancer-specific components in the total population of circulating vesicles. One of the first successful examples of this strategy was published in 2015 [17]. It was shown that an assessment of the panel of proteins and microRNA from plasma nanovesicles (exosomes) allowed for the accurate diagnosis of pancreatic cancer. Several subsequent reports confirmed the efficacy of this approach in different tumor entities, including cancers of the breast [18,19,20] and prostate [21,22] as examples. However, this second strategy assumes that cancer-related alterations are present in a considerable and easily detectable portion of circulating ENVs. The bias of this assumption was demonstrated by quantitative and stoichiometric analysis of the microRNA content in circulating exosomes [23]. Considering the well-known dependence of the plasma ENVs’ content on the method of their isolation [24,25], the analytic and diagnostic sensitivity turned out to be the most vulnerable aspect of this strategy. Thus, both approaches to ENV-based cancer diagnostics are not ideal, justifying the search for novel strategies. 

Colorectal cancer (CRC) is the third most common cause of cancer-related death [26]. It is associated with ineffective screening, late diagnosis, and increasing incidence in the younger population [27]. The active search for new diagnostic approaches for this cancer type was reflected in the extensive investigation of CRC-associated vesicular markers. The first significant study was published by Ogata-Kawata and co-authors [28], reporting a CRC-associated increase in several miRNA levels in the total population of circulating ENVs (exosomes). Subsequently, several investigations by others [29,30,31,32,33,34] and by us [35] explored the diagnostic potential of ENV-encapsulated miRNAs. However, the results of these studies are only weakly consistent. Considerable attention was paid to vesicular long noncoding RNA [36,37,38,39] and proteins [40]. For instance, the levels of the abovementioned Glipican-1 [41], extracellular matrix metalloprotease inducer CD147 [42], or the receptor FZD-10 involved in the Wnt signaling pathway [43] were shown to be elevated in the plasma ENVs of CRC patients. Despite the huge amount of research conducted, there is only a single clinical study (NCT04394572) to date aiming at the evaluation of the clinical utility of vesicular forms of integrins and metalloproteinases in CRC diagnostics. A recent review by Xiao and coauthors [44], providing a comprehensive analysis of the current status of ENV-based CRC marker research, indicated the ENV isolation issue, the great heterogeneity of the plasma ENVs, and the absence of universal CRC markers as the most important limitations in using ENVs as CRC markers. This fully reflects the current state-of-the-art of ENV-based liquid biopsy in cancer diagnosis in general.

In the present study, we explored a new ENV-based strategy in cancer diagnosis using the example of CRC. It was assumed (i) that cells of different tissues secrete extracellular vesicles, (ii) that these vesicles reflect the biochemical compositions of the parental cells, and (iii) that the tissue-derived ENVs reach circulation at amounts corresponding to the volumes and secretory activities of the respective source tissues. We hypothesize that the development of differentiated CRC (adenocarcinomas) might be associated with the elevated release of ENV-bearing markers of the intestinal differentiation of epithelial cells. Previously, we selected several intestine-specific, surface membrane-associated proteins and estimated their presence in the membrane of ENVs secreted by colon cancer cells in vitro [45]. In the present study, we proceeded towards the quantitation of ENVs showing such intestine-specific markers in the plasma of CRC patients and healthy donors using on-bead flow cytometry. On the basis of the results obtained, we can conclude that the quantification of tissue-specific ENVs in plasma may represent a new “liquid biopsy” approach to cancer diagnostics. 

## 2. Materials and Methods

### 2.1. Biological Material

The biological material was obtained from donors and patients treated at the N.N. Petrov National Medical Research Center of Oncology of Ministry of Health of Russia/NMRC of oncology (St. Petersburg) and the Ryzhikh National Medical Research Center of coloproctology/NMRC of coloproctology (Moscow). The venous blood was collected in vacutainers with EDTA, and the plasma was separated within 10 min after blood collection, was frozen, and was stored at −80 °C. Control blood samples were collected from healthy donors without any clinical signs of colorectal pathology at the blood transfusion department of NMRC of oncology. The plasma samples collected from patients were included in the study after histological confirmation (adenocarcinomas) of the diagnosis and disease staging. The characteristics of the patients are presented at Table 1.

The study protocol was approved by the ethics committee of the N.N. Petrov Research Institute of Oncology. All participants signed informed consent forms.

### 2.2. Reagents and Antibodies

Polyethylene glycol (PEG) 20 kDa (Serva, Heidelberg, Germany) and Dextran (DEX) 450–650 kDa (Sigma-Aldrich, Saint Louis, MO, USA) were used for ENV isolation. On-bead flow cytometry was performed using 4 µm latex particles from Exo-FACS kit (HansaBioMed Life Science, Tallinn, Estonia) or 1 µm streptavidin-coated super-paramagnetic particles/SPMP (Sileks, Moscow, Russia). The Biotin-conjugated antibodies against CLRN3 (CSB-PA818764LD01HU), GPA33 (CSB-PA857459ED01HU), and GCNT3 (CSB-PA009329LD01HU) were from Cusabio (China); the antibodies against PIG-Y (ABP56795), Reg IV (ABP56724), and DHRS11 (ABP56569) were from Abbkine (China); and the antibodies against Meprin A (LAA171Hu71), GAL4 (LAA304Hu71), Mucin 12 (LAL862Hu71), and PDCD6IP (PAB247Hu01) were from Cloud-Clone Corporation (China). The secondary antibodies against CD63 (FITC-conjugated, mouse monoclonal MEM-259) were from AbCam (USA), and that against CD9 (PerCP/Cyanine5.5-conjugated, mouse monoclonal HI9a) was from Biolegend (USA). Additionally, the 0.2% Tropix i-Block buffer (Thermo Fisher, Cleveland, OH, USA) was used.

### 2.3. Extracellular Nanovesicle (ENV) Isolation

The population of plasma ENVs enriched with exosomes was isolated with Plasma Two-Phase Polymer System (PTPS), as described recently [46] with minor modifications. Briefly, plasma samples were slowly thawed at 4 °C prior to removing the debris and large aggregates by centrifugation (300× *g* for 10 min, 2000× *g* for 10 min and 10,000× *g* for 10 min). The polymers were dissolved in 1.5 mL of the plasma at a concentration of 3.5% (PEG) and 1.5% (DEX) by vortexing for 1 h. In parallel, the same quantities of polymers were dissolved in 1.5 mL PBS for preparing the plasma protein-depleting solution (PDS). Two tubes containing the plasma and PDS were centrifuged at 1000× *g* for 10 min to speed up the separation of the polymer solutions into the lower phase (LP) and upper phase (UP). To deplete the LP from the plasma protein, the UP was carefully removed and replaced by PDS. The solutions were mixed and re-separated again. After separation, the second UP was discarded, and the second LP containing the ENVs was dissolved in PBS up to 100 µL.

### 2.4. Nanoparticles Tracking Analysis (NTA)

NTA measurements were performed using the Nanosight NS300 analyzer (Malvern Panalytical, UK). Each sample was studied in 4–5 different micro volumes by pumping the sample through a chamber. The duration of each measurement was 30 s (camera level: 14; shutter slider: 1259; slider gain: 366; and threshold level: for draft—5, for VF2—6). The experimental data were analyzed using Nanosight NTA 3.2 Software. 

### 2.5. Transmission Cryo-Electron Microscopy (Cryo-TEM)

Analyses were performed using a Jeol JEM-2100 microscope at Research Resource Center for molecular and cell technologies of St. Petersburg State University. Samples of the ENVs at a concentration of 7 × 1011 particles/mL were deposited on a carbon-coated copper mesh/Lacey Carbon Supported Copper Grids, size 50 nm (Sigma-Aldrich, USA). Excess samples were removed with a filter paper. Then, the sample was immersed in liquid ethane for rapid freezing and transferred to a cryostat for subsequent analysis by cryo-microscope.

### 2.6. Analysis of Total ENVs Population by Flow Cytometry

Exosomal markers (CD63 and CD9) on the surface of PTPS-isolated ENVs were assayed using the Exo-FACS kit (HansaBioMed Life Science, Estonia) according to the manufacturer’s protocol. ENVs immobilized on the surface of the latex beads were labeled with monoclonal antibodies to tetraspanins CD63 or CD9 and conjugated with FITC or PerCP-Cy5.5, respectively. Flow cytometry data were obtained on a CytoFLEX analyzer (Beckman Coulter, Brea, CA, USA) equipped for multi-parametric and multicolor analysis, including a 488-nm argon laser for measurement of forward light scatter (FSC) and orthogonal scatter (SSC). The complexes assembled without ENVs were used as a negative control. The data were analyzed with CytExpert Acquisition and Analysis Software Version 2.4. (Beckman Coulter, USA) software. After acquisition, the data were exported and analyzed using FlowJo version 10.1r5 (Treestar, San Carlos, CA, USA).

### 2.7. Preparation of Immune-SPMP and Analysis of ENVs Isolated by Immunosorption

The biotinylated antibodies were adjusted to a concentration of 10 ng/mL. Streptavidin-coated SPMP (1 mg/mL) were pretreated according to the manufacturer’s recommendation. The 1 µL of pretreated SPMP was added to the antibody solution (10 µL), gently mixed, and incubated for 1 h at 4 °C. Assembled immunoparticles (SPMP-AB) were washed with PBS from unbound antibodies, mixed with the ENVs (10–12 × 109 vesicles per reaction) in a volume of 100 µL PBS, and incubated with slow rotation overnight at 4 °C. The resulting complexes (SPMP-AB-ENVs) were washed three times with PBS, blocked in 200 µL 0.2% Tropix i-Block buffer for 1 h at 4 °C, and washed again with PBS. To quantify the assembled ENVs, the complexes were incubated with antibodies against the classic “exosomal” markers CD63-FITC and CD9-PerCP-Cy5.5 for 2 h at 4 °C in the dark. The resulting complexes were washed twice, diluted in 100 µL PBS, and analyzed by flow cytometry. As negative control, the SPMP-AB complex without ENVs underwent the same blocking procedures, labeling with CD63-FITC and CD9-PerCP-Cy5.5 antibodies, and washing steps. 

### 2.8. EV-Track 

All procedures used for ENV isolation and analysis were validated by EV-Track, a tool developed and supported by ISEV to enhance the transparency and interpretation of EV experiments (https://evtrack.org). The EV-TRACK ID is EV210142, 8 April 2021. 

### 2.9. Statistical Data Analysis

Illustration and statistical calculations were performed using the Nanosight NTA Software 3.2, CytExpert Acquisition and Analysis Software 2.4., Graph Pad Prism 6, Sigma Plot 12 and BioRender. A nonparametric Mann–Whitney U-test was used to evaluate the statistical significance of differences between the sample groups. The ROC (receiver operating characteristic) algorithm was used to evaluate the diagnostic potency of the markers. Significance levels were defined as * *p* < 0.05, ** *p* < 0.005.

## 3. Results

### 3.1. Isolation and Analysis of the Total ENV Population

The Plasma Two-Phase Polymer System (PTPS) method was applied for the isolation of the exosome-enriched population of plasma ENVs. In all cases, ENVs were isolated from 1.5 mL plasma and dissolved in 100 μL PBS. The size distribution and concentration of the isolated particles were determined by NTA after diluting 1:1000. The size of isolated nanoparticles showed a unimodal distribution, with a major fraction at 85–110 nm. The concentration of isolated nanoparticles varied in the range of 2–10 × 10^11^/mL. A representative example of the NTA results is shown in Figure 1A. The vesicular structure surrounded by the membrane was visualized by Cryo-TEM (Figure 1B). The surface membrane of isolated nanovesicles contained both common “exosomal” markers CD63 and CD9, as confirmed by on-bead flow cytometry (Figure 1C). Other characteristics of plasma nanovesicles isolated by PTPS have been described previously in greater detail [46]. Based on the results obtained, we concluded that the isolated population of ENVs consists mainly, but not exclusively, of exosomes. 

### 3.2. Selection of Potential Surface Markers of Colon Epithelium-Derived ENVs

We hypothesized that the development of adenocarcinomas from colon epithelium is associated with an increase in the concentration of colon epithelium-derived ENVs circulating in the plasma. To develop a method for the quantification of these vesicles, we selected potential surface markers using available databases [47,48,49,50]. We chose 16 proteins that are expressed exclusively or predominantly in colon/intestinal epithelium cells; have well-established cell surface membrane localization; and are preferentially detected as components of extracellular vesicles: CLRN3, GPA33, GCNT3, PIGY, REG4, MEP1A, LGALS4, Mucin 12, PDCD6IP, DHRS11, CD47, VAMP1, CEACAM5, CD177, CDH17, and CDH5. In our previous study, we had confirmed the expression of these proteins in the normal colon epithelium, their membrane localization, and vesicular expression by cultured colon cancer cells [45].

### 3.3. Quantification of Colon-Specific ENVs in Plasma of CRC Patients

To quantify the colon-specific fraction of plasma ENVs, we applied a method of immune-sorption followed by quantitative analysis. The principle of this approach has been described previously [15] and is presented schematically in Figure 2.

We used superparamagnetic particles (SPMP) coated with streptavidin and biotin-labelled antibodies (AB) against the selected potential colon-specific surface markers. The assembled immunoparticles (SPMP-AB) were incubated with a total population of plasma ENVs isolated by PTPS. Bound ENVs were labelled by incubating with antibodies against common exosomal markers (tetraspanins CD63 and CD9) and quantified by flow cytometry. To minimize a potential bias from uncertain levels of exosomal marker expression, each sample of SPMP-AB-ENV complexes was stained with CD63-FITC and CD9-PerCP antibodies and assayed in two channels in parallel. The results from a representative example of such an experiment are presented in Figure 3A.

First, immunoparticles SPMP-AB, containing antibodies against metalloproteinase Meprin A, were blocked with the Tropix i-Block buffer, stained with FITC-labeled anti-CD63 antibodies, and assayed as a negative control (ENV(−) control). This level of intensity was used as a signal background level. In parallel, SPMP-AB complexes were blocked, incubated with ENVs, and then labeled with FITC-coupled anti-CD63 antibodies. The panels in Figure 3A present the results of the analysis of ENVs isolated from a healthy donor (3.92% of positive particles) and CRC patient (21.41% of positive particles). The construction of pseudo-color/smooth plots (CD63 FITC versus FSC-H (forward Light Scatter–Height) allowed us to detail the distribution of the fluorescent intensity among the positive events: SPMP-AB-MeprinA(+)ENVs-CD63AB-FITC antibody complexes. The red regions reflect the amount of highly fluorescent beads, and the green and blue regions reflect the distribution of beads with middle and slight fluorescent signal correspondingly. As shown in the colored inserts, in the CRC sample, red and green smooths are shifted to the right site. This reflects a higher intensity of the fluorescent signal from the counted beads and, hence, a higher number of bead-attached MeprinA(+)ENVs.

The results of the analysis of the two groups of samples obtained from healthy donors (n = 20) and CRC patients with stage IIIb-IV of the disease (n = 20) are summarized in Figure 3B. Shown are the quantification results for Meprin A(+)ENVs captured by immunobeads SPMP-AB and labeled with anti-CD63 antibodies (FITC signal, left) or anti-CD9 antibodies labeled with PerCP (right). In both cases of labeling, the amount of Meprin A(+)ENVs was more variable and higher in the group of CRC samples vs. samples from healthy donors, and in the case of anti-CD9 antibodies used for labeling, this difference reached statistical significance. Next, we evaluated the diagnostic potency of these complex markers (SPMP-AB-ENVs-CD63-FITC and SPMP-AB-ENVs-CD9-PerCP) by ROC analysis. We obtained similar values for the area under the curve (AUC): 0.81 and 0.79, which integrally reflected the diagnostic specificity and sensitivity of this method (Figure 3C). 

Table 2 contains the results of the analysis of the ten best out of the sixteen markers included in the study; the complete data are presented in Appendix A. 

Again, the two groups of samples (healthy donors; n = 20) and CRC stage IIIb–IV patients; n = 20) were compared. Six other markers did not reveal any difference between the compared groups and were excluded from the further experiments. Table 2 gives relevant characteristics of the analyzed proteins retrieved from an open source, including scores reflecting colon epithelium-specific expression and cell surface membrane localization. Most of these proteins have already been reported as exosome-associated and were included in the ExoCarta database. The right part of Table 2 contains the results from our experiments. Colon epithelium-specific ENVs captured by different types of immunobeads were quantified after labeling with either anti-CD63-FITC or anti-CD9-PerCP antibodies. In each case, the percentage of positive immunobeads averaged for the two groups (CRC patients and healthy donors), the statistical significances of the comparison of the two groups assayed by Mann–Whitney test, and the AUC values estimated by ROC analysis are presented. Using all ten markers for the immobilization of colon-specific ENVs, an elevated level of these vesicles was detected in CRC samples. These results support the main hypothesis of this study: the development of differentiated adenocarcinomas of colon is associated with an increase in colon epithelium-specific ENVs circulating in the plasma. However, only in some cases did the observed differences between CRC and control groups reach statistical significance. This was true for REG4(+), MEP1A(+), and Mucin12(+) ENVs labeled with anti-CD9-PerCP antibodies as well as for CLRN3(+), GCNT3(+), PIGY(+), REG4(+), and Mucin12(+) ENVs labeled with anti-CD63-FITC. Thus, the labelling of immuno-captured ENVs with CD63-FITC seemed to be more effective compared to anti-CD9-PerCP. Interestingly, the AUC values were always higher (except for REG4(+)ENVs) after labeling with anti-CD63-FITC compared with anti-CD9-PerCP. This result indicates the influence of the selected method of immune detection on the analysis results, and the diagnostic potency of the method in general.

### 3.4. Multiplex Assessment of Colon-Specific ENVs in Plasma of CRC Patient

Having obtained promising results for the quantitation of colon-specific ENVs upon binding to the beads based on the separate markers, we carried out an experiment on their multiplex isolation and quantification. For this, we prepared immunoparticles consisting of SPMP and five biotin-labeled antibodies specific for CLRN3, GPA33, GCNT3, PIGY, and REG4 in equivalent amounts. The subsequent stages of the procedure of colon-specific ENV isolation were the same as described above. Since the labeling of immune-captured ENVs was more efficient when using anti-CD63-FITC antibodies, this method of labeling was used for multiplex analysis. In the final experiment, we aimed to evaluate the early diagnostic potency of the developed approach. Therefore, we used the plasma samples collected from CRC patients with an earlier stage of disease, i.e., stages II-IIIa (n = 28). A new group of healthy donors (n = 30) was used as the control. 

The results are presented in Figure 4. The negative control (immune-complex SPMP-5AB blocked with Tropix i-Block buffer and stained with CD63-FITC antibodies), and the representative results of colon-specific ENV quantification in plasma samples obtained from healthy donors or CRC patients, respectively, are shown in Figure 4A. Again, the colored inserts revealed increased fluorescent signal from the beads with a higher number of attached tissue-specific ENVs in the CRC sample. Averaging the results from the clinical groups revealed only a minor difference (4.198% of positive beads in CRC group vs. 2.057% of positive beads in the control group); however, the observed difference was statistically significant (*p* < 0.005; Figure 4B). The ROC analysis allowed us to estimate the diagnostic potency of the method, with the AUC determined at 0.82. With a cut-off level of 3.6% of positive immune-beads assayed by flow cytometry, the diagnostic specificity of our analysis reached 0.82 whereas the sensitivity reached 0.75. Taken together, our results confirm a CRC-associated increase in colon-derived ENVs in plasma. The proposed approach may thus be considered a promising option for ENV-based cancer diagnostics. 

## 4. Discussion

The analysis of tissue-specific ENVs has been explored as a method for noninvasively monitoring the acute rejection of transplants [52,53]. To the best of our knowledge, this is the first study on the exploration of tissue-specific ENVs as a disease marker. While our results indicate the potency of this approach for cancer diagnostic, several aspects require further research. The most general concerns are related to the still poor understanding of the heterogeneity of plasma ENVs and to the availability of appropriate tissue-specific markers. The malignant transformation and uncontrolled proliferation of certain epithelia may lead to an increase in tissue-specific ENV secretion and thus higher levels in plasma. While this was indeed seen in our study, it requires further confirmation in other tumor entities. To further explore and evaluate the performance of the diagnostic approach proposed here, the tissue-specific ENVs circulating in plasma must be precisely profiled in the healthy condition, thus requiring accurate reference values. This task requires profound knowledge of the specific biochemical composition of ENVs secreted by the selected epithelial cells of interest in vivo. The analysis of vesicles secreted by cultured cells may not be an appropriate approach, and for example, the newly proposed technology of ENV isolation from intestinal fluid may be a better method [54].

As soon as the biochemical specificity and physiological amounts of tissue-specific ENVs in plasma are defined, the link between malignant transformation of a given epithelium and profiles of tissue-specific ENVs in the circulation can be studied in detail. It must be presumed that histological variants, differentiation stages, and other tumor characteristics strongly influence the composition and number of vesicles released by tumor cells. This may also include the possibility that tumors such as anaplastic carcinomas secrete ENVs lacking any tissue-specific markers. This needs to be carefully excluded since, in this case, the proposed diagnostic approach would not yield accurate results. Thus, further investigation of the link between tumor cell biology and the biochemistry of the secreted ENVs is required to ensure sufficient diagnostic specificity with the proposed strategy. 

Our final comment is related to sensitivity. The method explored in our study allowed us to detect 1–4% of positive SPMBs with immune-captured ENVs, labelled by fluorescent antibodies, in the plasma of healthy donors. In the CRC samples, these parameters generally increased to 3–7%. These rather small differences between positive and negative values may suggest a relatively low diagnostic sensitivity of the method and a high risk of errors. Notably, however, some of these differences were still found to be statistically significant, and the statistical significance increased when switching to the multiplex analysis mode. Perhaps not surprisingly, this indicates the advantage of using multiple antibodies for ENV capture. Other, perhaps more efficient methods of ENV labeling, e.g., using lipophilic dyes [55], may help to further improve sensitivity.

As we presented a pilot study demonstrated promising results, we plan to expand this research in two directions: fundamentally and methodologically. On one hand, a deep analysis of plasma ENV compositions should be conducted to evaluate the main tissue sources of vesicles and the physiological ratio of different vesicular populations. We plan to use new methods of single-vesicle analysis such as stochastic optical reconstruction microscopy (STORM) [56] and fluorescently labelled ligands to membrane proteins specific to different types of epithelium. On the other hand, the analytic sensitivity of the method should be optimized to ensure accurate measurement of nonsignificant changes in the concentration of a specific vesicular population. This issue has several potential resolutions. First, the sensitivity of bead-assisted flow cytometry can be greatly improved by staining the plasma with lipophilic dye followed by size-exclusive chromatography. As we demonstrated recently [57], such an approach allowed us to detect a very small vesicular population. Second, alternative methods of nanozyme-based sensing [58] or plasmon resonance [59] can provide a higher sensitivity than flow cytometry and can by optimized for clinical application. Thus, after deepening the fundamental understanding of the pattern of circulating ENVs and optimization of the technology, we plan to evaluate the proposed approach using more representative groups of CRC patients to evaluate standard parameters of diagnostic potency (specificity, sensitivity, PPV, NPV, and accuracy) and to identify appropriate clinical application of the proposed method. 

## Figures and Tables

**Figure 1 cancers-13-03905-f001:**
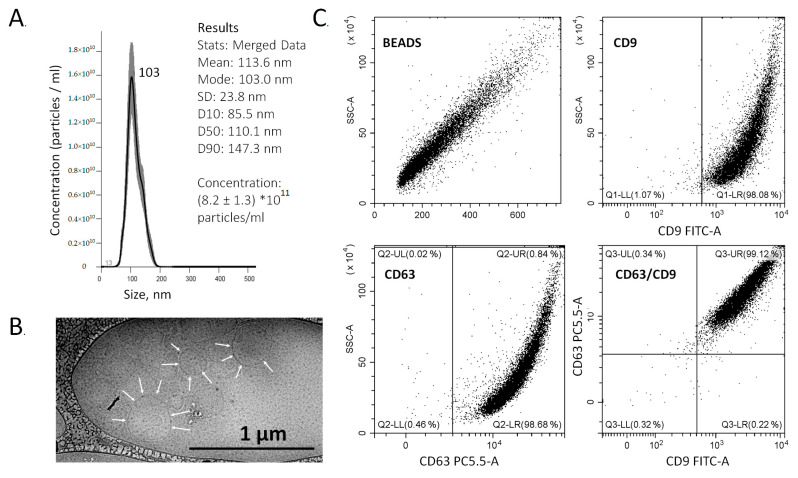
Characteristics of plasma ENV isolated by PTPS. (**A**) Evaluation of the size and concentration of ENVs using nanoparticle tracking analysis (NTA). (**B**) Morphology of particles imaged by Cryo-EM. Vesicular membranes are indicated by arrows. (**C**) Analysis of the expression of exosomal markers CD63 and CD9 on the vesicle membrane. Vesicles were nonspecifically bound onto the latex beads. After incubation with fluorescently labeled antibodies (CD9 or CD63), fluorescence intensities were determined for the corresponding channels (FITC and PC5.5). The lower right square is the result of combining the data from two channels and indicates that 99.1% of particles with fixed ENVs bind both antibodies, i.e., almost all vesicles are positive for both exosomal markers.

**Figure 2 cancers-13-03905-f002:**
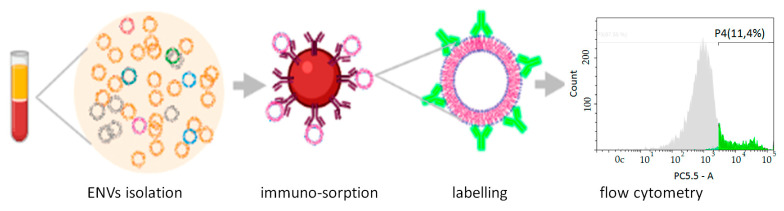
Workflow of colon-derived ENV isolation and analysis.

**Figure 3 cancers-13-03905-f003:**
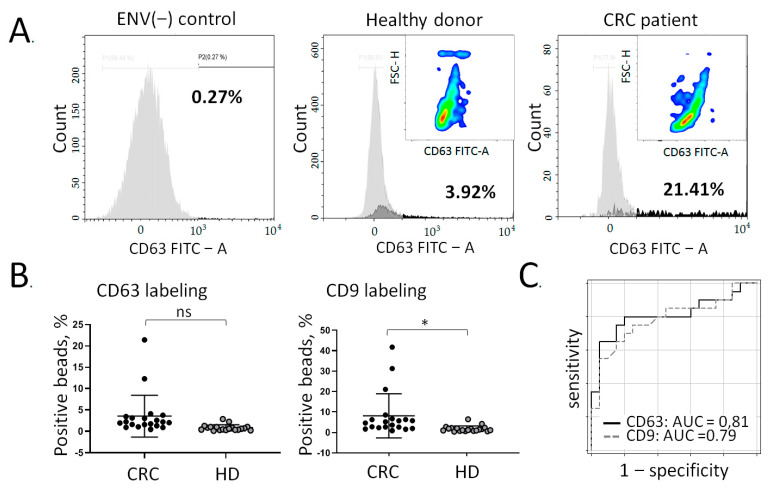
Quantitative analysis of MeprinA(+)ENVs in the plasma of CRC(IIIb-IV) patients and healthy donors. The total population of ENVs was isolated with PTPS, and the fraction of Meprin(+)ENVs was isolated using immune-beads decorated with antibodies against Meprin A. Captured MeprinA(+)ENVs were labeled with antibodies against the exosomal markers CD63 and CD9 and quantified by flow cytometry. (**A**) Representative examples of flow cytometry histograms after labelling with anti-CD63 antibodies (FITC signals). The colored inserts present pseudo-color/smooth plots of positive events. Lower panels: higher magnification. (**B**) Group comparison of CRC (IIIb-IV) patients (n = 20) vs. healthy donors (n = 20) after labelling with anti-CD63 or anti-CD9 antibodies. The statistical significance was estimated using nonparametric Mann-Whitney test (ns > 0.05; * *p* < 0.05). (**C**) Evaluation of the diagnostic potency of this method by receiver operating characteristic (ROC) analysis in the same group of samples.

**Figure 4 cancers-13-03905-f004:**
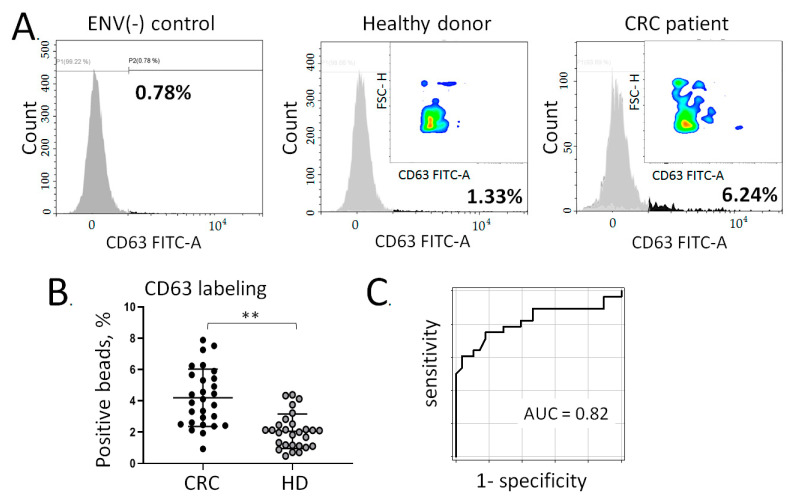
Multiplexed isolation and quantification of colon-derived ENVs in the plasma of CRC (II-IIIa) patients and healthy donors. The total population of plasma ENVs was isolated with PTPS, and the fraction of colon-derived ENVs was isolated using immunobeads decorated with antibodies against CLRN3, GCNT3, GPA33, PIGY, and REG4. ENVs captured by the immune beads were labelled with FITC-conjugated antibodies against the exosomal marker CD63 and quantified by flow cytometry. (**A**) Representative examples of the histograms (negative control, healthy donor, and patients) after labelling with anti-CD63-FITC. Colored inserts present pseudo-color/smooth plots of positive events. (**B**) Direct comparison of the CRC (II-IIIa) patients (n = 28) vs. the healthy donors (n = 30). The statistical significance was estimated using nonparametric Mann-Whitney test (** *p* < 0.005). (**C**) Evaluation of the diagnostic potency of the method by receiver operating characteristic (ROC) analysis in the same groups of samples.

**Table 1 cancers-13-03905-t001:** Clinical characteristics.

	Healthy Donors (HD)	CRC Patients, IIIb–IV	CRC Patients, II–IIIa
Number	50	20	28
Age, mean (SD)	50 (6.3)	53 (9.3)	51 (5.5)
Gender, (m/f)	36/14	13/17	15/13

**Table 2 cancers-13-03905-t002:** List of colon epithelium-specific protein surface markers.

Protein Name	UniProt ID	Colon-Specific Expression ^1^	Surface Membrane Localization ^2^	Exo Carta ID	CD9				CD63			
					CRC Patients	Health Donors	CRC vs. HD	AUC	CRC Patients	Health Donors	CRC vs. HD	AUC
CLRN3	Q8NCR9	27.7	***	119467	7.71	2.8	ns	0.76	3.2	1.4	**	0.78
GPA33	Q99795	46.9	*****	10223	4.8	4.1	ns	0.54	2.1	1.5	ns	0.67
GCNT3	O95395	37.6	****	9245	6.4	2.5	ns	0.64	2.8	1.0	*	0.71
PIGY	Q3MUY2	10.6	*****	84992	7.2	3.3	ns	0.66	3.2	1.3	**	0.78
REG4	Q9BYZ8	55.5	***	83998	7.5	2.2	*	0.78	3.0	1.0	*	0.74
MEP1A	Q16819	100.5	*****	4224	8.1	2.8	*	0.79	3.6	1.5	ns	0.81
LGALS4	P56470	190.6	****	3960	5.6	2.4	ns	0.69	2.2	1.2	ns	0.71
Mucin 12	Q9UKN1	25.2	*****	-	7.3	2.7	*	0.75	3.0	1.2	*	0.78
PDCD6IP	Q8WUM4	36.8	*****	10015	5.7	4.1	ns	0.6	2.5	1.7	ns	0.66
DHRS11	Q6UWP2	38.7	**	360583	7.7	3.2	ns	0.63	2.7	1.4	ns	0.68

^1^ Fold of increase in Normalized eXpression (NX) levels for specific proteins in the colon epithelium vs. any other tissue region/region/cell type (based on The Protein Atlas [47]). ^2^ The surface membrane localization is derived from database annotations, automatic text mining of the biomedical literature, and sequence-based predictions. The confidence of each association is signified by stars, where ***** is the highest confidence and * is the lowest (based on subcellular database compartments [50,51]).

## Data Availability

The results are either presented in the article or can be provided upon request.

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
