# Peer review of "Evaluation of Colon-Specific Plasma Nanovesicles as New Markers of Colorectal Cancer"

_cancers, 2021, doi:10.3390/cancers13153905_

Round 1

Reviewer 1 Report

Inga et al. investigated the potential use of colon-specific plasma nanovesicles for early-stage colon cancer diagnosis.

The authors relied on the literature and their earlier publications to choose and validate 16 cell surface proteins for their potential to be biomarkers on epithelium derived ENVs. Their multiplexed colon-specific ENVs relying on CLRN3, GPA33, GCNT3, PIGY and REG4 might have early stage diagnostic value. The results presented here are interesting for the field and might be a good starting point for future research that aims to establish early-stage biomarkers for colon cancer.

Minor points:

  1. The authors stated that they have tested 16 proteins but only list the ten best out of this in Table 2. Listing the details of the remaining 6 proteins would be useful for future reference.
  2. Figure 4B compares the multiplexed assay results for CRC and health donors (HD). The text says the CRC group had 4.198% positive beads rate while the control group had 3.153%. However, the mean value for the HD sample on the graph is below 3%. What is the reason of this discrepancy?
  3. Minor grammar mistakes and typos must be corrected. For example, GraphPad Prism is written with an ‘s’ not with a ‘z’.

Author Response

Dear reviewer,

We'd like to thank you for your careful review of the manuscript.  We agree with all their comments, and we have revised our manuscript accordingly. We respond below in detail to each of the comments.

Sincerely, the authors of the manuscript

Reviewer 2 Report

the current study is a novel approach to the concept of Liquid biopsy in colon  cancer

the major problem is the small number of cases included in the study

the authors should extent the discussion and try to put the study into perspectives, paying attention in the limitations of their research

High novelty but major changes are warranted

Author Response

Dear reviewer,

we'd like to thank you for your careful review of the manuscript.

Sincerely, the authors of the manuscript

Round 2

Reviewer 2 Report

no additional comments